# Bone Remodeling during Orthodontic Movement of Lower Incisors—Narrative Review

**DOI:** 10.3390/ijerph192215002

**Published:** 2022-11-15

**Authors:** Edyta Kalina, Anna Grzebyta, Małgorzata Zadurska

**Affiliations:** Department of Orthodontics, Medical University of Warsaw, Stanislawa Binieckiego 6, 02-097 Warsaw, Poland

**Keywords:** bone remodeling, incisor inclination, orthodontic treatment, gingival recession, bone fenestration, bone dehiscence

## Abstract

The tooth movement in the alveolus is possible due to bone remodeling. This process could be the risk factor for the formation of gingival recessions—the most common side effects of orthodontic therapy. Gingival recessions are found 5.8–11.5% more frequently among the orthodontically treated patients. What is more, anterior mandibular teeth are the ones most prone to gingival recession dehiscences and fenestrations. The aim of this narrative review was to evaluate, based on CBCT (Cone beam computed tomography) scans, the changes in the alveolar bone of lower incisors in adolescent and adult patients after orthodontic tooth movements. From the pool of 108 publications, a total of 15 fulfilled the criteria of this review. Both retrospective and prospective longitudinal studies—using CBCT or CT (Computed Topography) and evaluating alveolar bone changes in mandibular incisors during orthodontic treatment performed before and after teeth movement—were included. In the group of growing patients, either proclination or retroclination of mandibular incisors led to increase of the distance from CEJ (cementoenamel junction) to marginal bone crest. The difference in bone loss was greater on the lingual side of the incisors in both types of tooth movement. The results were similar for adults patients. The thickness of the alveolar bone was reduced after proclination (total bone thickness) among growing and non-growing patients and retraction (lingual and buccal) of lower anterior teeth in the group of growing patients. The only improvement was measured for buccal thickness of mandibular incisor in bimaxillary protrusion patients treated with extraction therapy. The control of retraction movement (more root than crown movement) enhanced preservation on bone height and thickness. In order to minimize possible deterioration and place teeth in the center of alveolus, CBCT monitoring and scrupulous clinical evaluation are recommended.

## 1. Introduction

The frequency of malocclusion is evaluated to be approximately 39.5–76% [1,2,3] in adult patients, comprising 74.7% of class I, 19.56% of class II and 5.93% of class III malocclusion worldwide. In the group of mixed dentition patients, distribution of these malocclusions was 73%, 23% and 4%, respectively [4]. High demand for orthodontic treatment exists not only among patients—professional dentists perceive orthodontic therapy as a significant part of interdisciplinary dental treatment. The aim of orthodontic treatment is not only to improve dental esthetics and function or to provide optimal occlusion but also to preserve or improve the condition of periodontal tissue. However, in the group of orthodontically treated patients, the frequency of occurrence of gingival recessions or bone dehiscences is higher than in the general population. Gingival recessions are found 5.8–11.5% more frequently in the orthodontically treated group, and also anterior mandibular teeth are the ones most prone to gingival recession dehiscences and fenestrations [5,6,7,8].

Application of orthodontic forces induces relaxation of periodontal ligaments in the pressure area and elongation of periodontal ligaments in the tension zone. The chain of biologic reactions activated by mechanical stimulus provokes remodeling of the alveolar socket [9]. The theory assumes that aseptic necrosis occurs in the compression area, and bone apposition—on the opposite side. In this scenario, proclination of incisors would be the causative factor of bone dehiscences, whereas retroclination or retraction of anterior teeth would result in buccal bone apposition and a decrease in bone height on the palatal or lingual side. In fact, non-linear structure of periodontal fibers and differences in bone density and metabolic rate, as well as direction and magnitude of orthodontic force, determine the type of bone reaction [10].

The limit of orthodontic tooth movement is the cortical plate. Violating the bone envelope can lead to such adverse reactions as bone dehiscence, gingival recession and root resorption. Anatomically, the alveolar bone becomes thinner from the posterior to the anterior region in the mandible. The thickness of the buccal bone of lower central incisors is approximately 0.07–0.17 mm at cervical level, 0.48–0.64 at mid-root level and approximately 2.37–3.71 mm at apex level [11]. Therefore, in the area of the mandibular symphysis, the direction and amount of tooth movement should be subject to thorough planning.

Findings from studies conducted on lateral cephalometric images are not conclusive. Authors reported a decrease of anterior alveolar bone width and height after retraction of upper and lower incisors [12]. On the other hand, it was demonstrated that, under some circumstances, the increase in labial alveolar bone due to orthodontic mandibular incisor retraction can be expected [13]. However, measurements on 2D teleoroentgenograms can be deceptive because of their limitations: overlapping of structures on the right- and left-hand side, with image enlargement and deformation due to changes in the head position during image acquisition [14,15,16,17]. Also, Fuhrmann proved that 2D cephalometric images overestimated the quantity of labial and palatal bone plate [18,19], thus they are not appropriate tools for the assessment of alveolar bone condition before and after orthodontic treatment.

The advent of cone-beam computed tomography (CBCT) has made it possible to evaluate the height and thickness of the alveolar bone and to evaluate the change of position of every single tooth. Kalina et al. proved, basing on CBCT, that the orthodontic treatment can be carried out without posing a high risk of gingival recession if the periodontal biotype is respected [20]. 

The aim of the study was to present up-to-date research indicating the changes in the alveolar bone of lower incisors, in developing and adult patients, after orthodontic tooth movements based on CBCT scans.

## 2. Materials and Methods

A search of literature was conducted in July 2022 using four databases: PubMed/Medline, Web of Science, Cochrane and Scopus. Both retrospective and prospective longitudinal studies—using CBCT or CT (Computed Topography) and evaluating alveolar bone changes in mandibular incisors during orthodontic treatment performed before and after teeth movement—were included. Reviews, cross-sectional studies, case reports and animal studies were excluded. All selected articles were in English.

## 3. Results

One hundred and eight studies were identified after the initial screening of titles and abstracts. Subsequently, duplicates, abstracts, animal studies, in vitro studies, studies based on lateral cephalometric images and manuscripts published in languages other than English were excluded. Two studies were excluded due to a wide range of patient age (10–over 40) in the study groups [21,22]. After full-text reading, 15 studies, which reported on bone remodeling after orthodontic movement of mandibular incisors, were included. Main outcomes of the analyzed studies are included in Table 1.

### 3.1. Incisor Proclination in Growing Patients

Matsumoto et al. reported that lower incisor proclination in young patients (mean age 11.23 y. o.) led to a reduction of alveolar bone in terms of height and thickness [23]. Moreover, authors determined a threshold beyond which tooth movement rapidly increased bone loss, i.e., a change of L1-NB (angle between lower incisor to Nasion-B Point line), a change of 0.71 mm and an IMPA (incisor mandibular plane angle) change of 3.02°. Logistic regression estimated a 50% probability of vertical bone loss at an L1-NB change of 2.00 mm or an IMPA change of 8.028.

Castro et al. conducted research on class I malocclusion patients with mild and moderate crowding treated without extraction. This study found that the distance from the cementoenamel junction to the bone crest increased in 57% of the cases, most often in the buccal (75%) and lingual (72%) surfaces of the mandibular central incisors. The difference in bone loss was greater on the lingual side of the incisors [24].

### 3.2. Incisor Retroclination in Growing Patients

In 2012, Lund et al. published a study on 152 patients aged 10–19 treated with extraction therapy. They found an increase in the distance from CEJ to marginal bone on the buccal aspect of lower central incisors in 67.7% and 74.6% of lateral incisors. The increase on the lingual side was 95.4% and 91.1%, respectively. The difference in distance greater than 2 mm occurred in 22% of central incisors, 24.1% of lateral incisors buccally, 83.4% of central incisors and 66% of lateral incisors lingually [37]. 

In two articles, authors used computed tomography (CT) obtained before, and three months after incisor retraction, to assess alveolar process changes [35,36]. Regarding the thickness of the bone, Sarikaya et al. reported a decrease in buccal bone at coronal level and in lingual site at coronal, mid-root and apical levels [35]. Similar results were obtained by Krishna et al., with the most significant bone width reduction detected at coronal level [36].

Maspero et al. investigated changes in the alveolar bone in all groups of upper and lower teeth in relation to inclination. Mean change of lower central and lateral inclination was −0.3° and −2.1°, respectively. Orthodontic treatment in the group of 11–16 year-old individuals class I with crowding patients caused buccal and lingual vertical and horizontal bone loss. Moreover, regression analysis indicated that torque changes of ±5 degrees seemed to produce an apical alveolar bone loss of up to −1.5 mm for the lingual sides in mandibular central incisors. Greater variability was found in the buccal aspect of central and lateral incisors, where alveolar bone thickness underwent up to 2.5 mm of bone loss and apposition of up to 0.5 mm [25]. 

### 3.3. Incisor Proclination in Non-Growing Patients

In patients with class I malocclusion, non-extraction therapy with slight proclination of lower incisors (mean change in IMPLA angle 4.02°) led to reduction in the width of alveolar bone 6 mm below CEJ, and in distance from CEJ to facial marginal bone. Authors also reported a decrease in alveolar ridge thickness and the interdental septum at mesial and distal sides, and demonstrated a correlation between the degree of crowding (irregularity index) and vertical bone loss [33].

In contrast, other study in the group of Class I and II patients with mild to moderate crowding did not demonstrate a reduction in total thickness of the alveolar ridge but rather a statistically significant labial cortical bone thinning of 3 mm, 6 mm and 9 mm from CEJ. There was no linear correlation between the cortical bone thickness change and IMPA changes [31].

Garlock et al. evaluated the remodeling of right central mandibular alveolar bone among 57 patients with class I or II malocclusion after orthodontic treatment with self-ligating brackets. Large variations in most variables were reported. Both bone gain and loss was detected in subjects, but on average, 1.12 mm of facial and 1.33 mm of lingual bone loss were reported, as well as 0.29 mm reduction in cortical lingual bone thickness at mid-root level. Additionally, authors indicated that thin mandibular symphysis and thin pretreatment cortical bone at apex level are risk factors for bone dehiscences. They found no correlation between the magnitude of tooth proclination and bone loss [32].

High-angle, class III patients are characterized by extremely thin alveolar bone in the anterior part of the mandible. During the presurgical phase of treatment, Ma et al. reported 1.57 mm of buccal and 2.82 mm of lingual vertical bone loss, as well as 7.11 ± 5.16 mm^2^ alveolar bone area loss on the lingual side [34].

In 2020, Yao et al. evaluated changes in alveolar bone after presurgical orthodontic treatment in a group of 29 class III adult patients with different facial divergence. Mandibular incisors proclination induced a decrease in both buccal and lingual bone height and a decrease in bone thickness at apex level, irrespective of facial growth pattern. Authors indicated factors affecting bone changes: facial divergence (MPA angle), tooth axis proclination, treatment time, tooth site and irregularity index [26]. On the contrary, Lee et al. disproved the correlation between the degree of incisors inclination and the extent of alveolar bone change. [28]

The results of Yao’s study [26] are partially in consent with previous studies by Sun et al. [27] and Lee et al. [28], where bone margin of labial and lingual bone was also lowered after presurgical orthodontic treatment in class III patients; however, the total apical bone thickness did not change.

### 3.4. Incisor Retroclination in Non-Growing Patients

In two studies, incisor retraction after premolar extraction in class I bialveolar protrusion patients induced changes in the shape of the alveolar mandibular bone [29,30]. Significant vertical bone loss occurred on the lingual side of teeth—2.73 mm in a study by Hung et al. [30] and 3.95 mm in a study by Zhang et al. [29]; the loss on the labial side was smaller but still statistically significant at 0.36 mm and 1.56 mm, respectively. In addition, the horizontal loss of lingual bone at five levels and buccal bone at crestal level was reported by Zhang [29]. On the contrary, Hung et al. observed an increase in measurement of buccal bone thickness. Moreover, they reported negative correlations between the range of root movement (apex displacement) and labial and buccal vertical bone loss and a significant positive correlation with buccal bone thickness [29].

## 4. Discussion

It is widely accepted that, whenever orthodontic tooth movement occurs, the bone around the alveolar socket remodels. On the other hand, orthodontic treatment is described as one of the etiological factors of dehiscence, fenestration and gingival recession. Is it possible to distinguish between a normal reaction of periodontal tissue and an iatrogenic effect? 

In the group of growing patients, either proclination or retroclination of mandibular incisors led to increase in the distance from CEJ to marginal bone crest [23,24,25,37]. What is more interesting, the thickness of the alveolar bone was reduced after proclination (total bone thickness) [23] and retraction (lingual and buccal) of lower anterior teeth [35,36]. The results were similar for adults patients [26,28,31,32,33,34]. The only “gain” in the measured variable was for buccal thickness of mandibular incisor in bimaxillary protrusion patients treated with extraction therapy [29,30]. The control of retraction movement (more root than crown movement) enhanced preservation on bone height and thickness [30]. The results of this narrative review confirm the statements from systematic review by Guo et al. They concluded that alveolar bone height and thickness, especially at the cervical level, decreased during both labial and lingual movement of anterior teeth [38].

It is commonly believed that the potential of bone to remodel is much greater in growing patients then in adult patients; therefore, orthodontic therapy in younger patients is safer with regard to periodontal tissue than in adult patients [22]. Surprisingly, the authors reported that the limits in forward movement of lower incisors might be less than previously expected. Exceeding the threshold of L1-NB change by more than 0.71 mm or an IMPA change of more than 3.028° may induce an adverse reaction in bone tissue [23]. Similarly, Maspero et al. indicate that movements exceeding +/−5° on lower central incisors are considered to be at risk of developing bone resorption at apical level [25]. On the other hand, extraction therapy and incisor retraction showed a mean 5.7 mm increase in distance between CEJ and MBC (margin bone crest) lingually, in some cases measured dehiscences were bigger than 8 mm [37]. Although during adolescence tooth eruption occurs, which may be the reason for a small increase in the CEJ-MBC distance, results of the above studies suggest that also in growing patients much attention should be paid during planning of antero-posterior incisor movements.

The limits for safe incisor movements in adult patients are not established, but the majority of authors underline that the prevalence of bone dehiscences, both before and after orthodontic treatment, is higher in older patients [22]. The degree of dental crowding was associated with the risk of bone dehiscence when treatment was performed without extraction [33]. The mean change in inclination 4.02–5.08 led to buccal bone height decrease [31,33]. 

In skeletal class III patients incisor decompensation before orthognathic surgery poses a high threat to their periodontal condition. The extent of bone loss is modified by: facial divergence, incisor irregularity, tooth site, treatment time and change in proclination [26]. Lee et al. explain the lack of correlation between the degree of incisor inclination and bone change by the influence of periodontal environment, the gingival type, patients’ oral habits and other factors [28].

Another aspect which should be considered is phenotype. Periodontal tissue in thin phenotype patients is characterized by thin alveolar bone and frequent dehiscences and fenestrations [39]. Those are more prone to recede when subjected to mechanical trauma, occlusal trauma, bacterial biofilm or orthodontic forces [40]. The thinner mandibular symphysis and pretreatment cortical bone, the greater the vertical bone loss during forward movement of incisors [32]. It is recommended to use CBCT in patients with thin phenotype to evaluate bone limits and plan proper teeth movements [20]. But the presence of dehiscences at the onset of studies did not predispose to larger bone resorption. In Matsumoto et al., the highest percentage of alveolar bone loss and the biggest difference in mean change in buccal bone height was in a group of patients with no dehiscences before orthodontic treatment [23]. However, it was also the group in which incisors were proclined in the highest degree. Also, another study did not find the correlation between the extent of pretreatment vestibular defect depth and the treatment-related change in this measure [37]. However, it is probable that orthodontists planned tooth movements with regard to 3D images, so the movements were limited when dehiscences were present originally.

Skeletal class III patients are a very special group because the alveolar bone around incisors is originally thinner than in class II or normal occlusion patients [41,42,43]. Moreover, incisors are subjected to significant forward movement before surgery. The bone loss, in terms of height and thickness, occurs not only before surgery [26,34] but continues until the completion of treatment [34]. Buccal thickness did not change in Yao et al. [26] and Lee et al. [28]; however, this can be explained by the measurement method used. 

In support of orthodontic treatment in patients with a high risk of gingival recession and bone dehiscences, the American Academy of Periodontology recommends phenotype modification via hard tissue augmentation with particulate bone grafting together with corticotomy [44]. In terms of soft tissue augmentation, the application of subepithelial connective tissue graft is regarded as the gold standard. Attempts have been made to search for alternative materials, such as collagen porcine dermal matrix [45], 24% EDTA (Ethylenediaminetetraacetic acid) and enamel matrix derivatives [46,47] and concentrated growth factors [48].

The point of application and the direction of force/a pair of forces determines the tooth movement. In three studies, correlation was found between proclination of incisors and facial bone recession [23,25,33]. Also, translational forward movement, bringing the apex closer to the facial cortical plate, induced a reduction in buccal bone height [32]. Two studies describe the tooth movement as controlled lingual tipping [35,36]. The retraction forces applied to incisors were concentrated at the alveolar crest, producing more evident alveolar bone loss at marginal and mid-root level. When the retraction occurred with more torque than tipping, the buccal thickness increased and marginal bone crest remained at higher levels compared to retroclination movement [30]. However, on the basis of a meta-analysis, bone remodeling is not stable after retraction. Both alveolar bone loss on the lingual side and alveolar bone gain on the labial side were obvious 1–3 months after retraction but less obvious after orthodontic treatment [38]. 

It is known that extrusion forces induce bone apposition under some circumstances [49], whereas tooth intrusion usually results in bone loss [50]. The vertical position of lower incisors was controlled only in Valerio et al., showing a correlation between extrusion (evaluated as an increase in the distance between pulp chamber ceiling and lingual plate) and reduced thickness in interdental septum, but no correlation with facial and lingual bone height [33]. 

The results of the above-mentioned study must be evaluated with caution, due to their heterogenicity and limits. Only two studies evaluated changes in bone dimensions in relation to bone morphology before treatment [23,26]. Few studies objectively measured the range of tooth movement [24,37]. Also, the change in inclination of incisors measured on lateral cephalometric images [23,27,34,35,36] did not provide information regarding changes for each incisor individually. Few studies evaluated the change in inclination for each tooth in CBCT or digital dental models [23,25,26,28,31,32,33]. Only in one study did the authors also assess vertical movements of incisors [33]. None of the studies took teeth rotation into account. Moreover, apart from one study [26], the authors did not consider differences in skeletal patterns in study groups. 

The analyzed studies were also not homogenous with regard to the timing of the second CBCT. Some of them were performed after orthodontic treatment [23,25,28,29,30,31,32,34,37] and some in the period after particulate tooth movements [26,27,33], whereas others—strictly after planned movements [35,36]. Wainwright found in his histological research that once the cortical plate had been penetrated, the buccal surface of the root was deprived of cortical bone. There was some osteogenesis after a four-month period of retention, but it was not sufficient to cover the root. Only after the tooth relapse did repair occur [51]. Some authors suggest that the cortical bone needs about six months to reestablish after pronounced tooth movement [52], whereas other neglect repair or remodeling even several years after treatment [53].

Last but not least, described studies most often used CBCTs with a voxel size of 0.23–0.377mm [23,24,25,26,29,31,34] or 0.15 mm [27]. Regarding the dimensions of buccal and lingual bone thickness in anterior region of the mandible, changes in these values after orthodontic treatment may not be detected [54,55,56]. It should be noted that the chance of a false positive detection of a dehiscence increases with larger voxel size. Only the study by Valerio et al. used CBCT with a 0.076 mm voxel size, which can provide better spatial revolution [33]. Future studies should weigh between patient exposure to radiation dose and optimal tissue imaging.

## 5. Conclusions

The bone remodeling process is essential to displace teeth. This is the reason for tissue dimension changes during orthodontic treatment. Uncontrolled movements beyond the initial bone limits might induce significant bone dehiscences, a risk factor for gingival recession. The etiology of dehiscences during orthodontic treatment is multifactorial and includes: direction, magnitude and duration of orthodontic forces, size and initial tooth position, alveolar bone anatomy, occlusal trauma, exposure to bacteria, oral habits and individual biological response to orthodontic forces. All of these factors should be controlled before and during orthodontic therapy. 

Periodontal accelerated osteogenic orthodontics (PAOO) is a clinical procedure that combines selective alveolar corticotomy, particulate bone grafting and the application of orthodontic forces. This procedure is theoretically based on the bone healing pattern known as the regional acceleratory phenomenon (RAP). With this technique, the preexisting alveolar volume does not have to be a limitation, and teeth can be moved two to three times further in [1/3] to [1/4] the time required for traditional orthodontic therapy [57,58]. According to Wilcko et al., it can be used to treat moderate to severe malocclusions in both adolescents and adults and can reduce the need for extractions. PAOO can also replace some orthognathic surgery. Special care should be given when planning treatment in skeletal class III patients. It is advisable to consider corticotomy or grafts in order to improve alveolar bone condition during presurgical incisor decompensation, but it will not replace the orthognathic surgery for severe cases. Extraction therapy, which is a method of choice in protrusion patients but also frequently proposed in cases of moderate and severe crowding, is safe for periodontal tissue only if incisor root displacement is meticulously controlled. In order to minimize possible deterioration and place teeth in the center of alveolus, careful diagnosis and monitoring with the use of CBCT is recommended. 

## Figures and Tables

**Table 1 ijerph-19-15002-t001:** Main outcomes of the studies analyzing changes in alveolar bone dimensions around mandibular incisors during orthodontic treatment.

Study Design	Author, Year of Publication	Patient Type/ Mean Age & Age Range (If Known)	Orthodontic Teeth Movement	Follow-Up Time	Alveolar Bone Changes	Other Outcomes
Retrospective						
Case control	Matsutmoto et al. [23]	48 skeletal cl. II	proclination	after treatment (mean treatment time: 28.0 months)	BH↓ TT↓ (5 mm and 15 mm from CEJ)	- alveolar buccal height was reduced most in patients with no dehiscences before treatment - 50% probability of vertical bone loss at an L1-NB change of 2.00 mm or, equivalently, an IMPA change of 8.02
		mean age: 11.2				
	Castro et al. [24]	30 skeletal cl. I patients	non-extraction treatment of mild to moderate crowding		BH↓ LH↓	- the change in vertical bone dimesnion cannot be contributed to orthodontic treatment
		mean age: 13.3				
	Maspero et al. [25]	22 cl. I patients	non-extraction treatment of mild to moderate crowding	after treatment (mean time: 22.0 ± 4.2 months)	BH↓ LH↓ BT↓(midrootapex level) LT↓ (midrootapex level)	- central incisors: torque changes of ±5 degrees - produce an apical alveolar bone loss up to −1.5 mm for the lingual sides and up to 2.5 mm of bone loss or apposition of up to 0.5 mm for the buccal sides - lateral incisors: apical facial bone remodeling ranging from −2 mm to +0.5 mm for torque variations between 0–4 degrees
		mean age: 13.0 (range: 11–16)				
	Yao et al. [26]	29 patients cl. III	presurgical treatment (lower incisor proclination)	after presurgical treatment (mean time: 11.0 months)	BH↓ LH↓ LT↓ TT↓	- factors of importance for bone changes: facial divergences, incisor inrregularity index, tooth site, treatment time, change in proclination
		mean age: 21.2				
	Sun et al. [27]	15 patients cl. III	presurgical treatment (lower incisor proclination)	after presurgical treatment (mean time: 11.8 months)	BH↓ LH↓ BT↑ LT↓	
		mean age: N.D. (range: 18+)				
	Lee et al. [28]	25 patients cl. III	presurgical treatment (lower incisors proclination)	after presurgical treatment (mean time: 21.4 months) and after debondind	BH↓ LH↓ BT↓	- no correlation between the degree of incisor inclination and the extent of alveolar bone change
		mean age: 26.3				
	Zhang et al. [29]	26 bimaxillary protrusion patients	retraction following premolars extraction	after treatment (mean time: 29.8 months)	BH↓ LH↓ BT↑ LT↓	
		mean age: 20.6 (range: 18–31)				
	Hung et al. [30]	24 bimaxillary protrusion patients	retraction following premolars extraction	after treatment (mean time: 31.2 months)	BH↓ LH↓ BT↑ LT↓	
		mean age: 19.3 (range: 11–27)				
Cross sectional study	Filipova et al. [31]	58 patients cl. I or cl. II	non-extraction treatment of mild to moderate crowding	after treatment	BT↓ (3, 6 and 9 mm from CEJ) TT→	- no correlation beteween change of inclination and change of buccal bone thickness - weak negative correlation between L1-Apog change and coritical bone thickness at 6 and 9 mm from CEJ level
		mean age: 23.2				
	Garlock et al. [32]	57 patients cl. I or cl. II	non-extraction treatment	after treatment (mean time: 22.7 ± 7.3 months)	BH↓ LH↓ TT→	- a thinner mandibular symphysis at the tooth apex was associated with an increase in facial vertical bone loss - thinner pretreatment cortical bone at the apex level was correlated with greater facial vertical bone loss - changes in IMPA were not correlated with facial vertical bone loss - thinning of cortical bone occurs on the surface undergoing vertical bone loss - movements of the mandibular incisor apex moving toward cortical bone produce greater amounts of vertical bone loss
		mean age: 18.7				
Cohort study	Valerio et al. [33]	32 patients cl. I	non-extraction treatment of anterior dental crowding	3–6 months after treatment (mean time: 17.4 ± 3.5 months)	BH↓ TT↓ (6mm from CEJ)	- irregularity index correlated with: changes in bone septum height on both the mesial and the distal side (strongly); changes in F-CEJ-MBC and L-CEJ-MBC (moderately and negatively) - increase in the IMPLA angle was correlated with a decrease of facial height of the alveolar ridge (from the facial marginal bone crest to the lingual plane)
		mean age: 23.0 (range: 18–29)				
Prospective						
Case control	Ma et al. [34]	30 patients cl. III high angle	presurgical treatment (lower incisors proclination)	after presurgical treatment and after debonding	BH↓ LH↓ BT↓ (8 mm from CEJ at midroot and apex level) LT↓ (4.6 mm from CEJ, at midroot and apex level) TT↓ (4.6 mm from CEJ, at midroot and apex level)	
		mean age: 20.9				
Cross sectional study	Sarikaya et al. [35]	19 bimaxillary protrusion patients	retraction after premolars extraction	3 months after retraction	BT↓ (3 mm from CEJ) LT↓ (3 mm and 6 mm from CEJ)	
		mean age: 14.1				
	Krishna et al. [36]	10 bimaxillary protrusion patients	retraction after premolars extraction	3 months after retraction	BT↓ (3 mm from CEJ) LT↓	- the changes differed depending on particular incisor (lateral/central, left/right)
		mean age: 15.0				
Cohort study	Lund et al. [37]	152 patients cl. I	retraction after premolars extraction	after treatment (mean treatment time 20.7 months)	BH↓ LH↓	
		mean age: 15.3 (range: 10–19)				

Abbreviations: cl.II—class II malocclusion, BH—buccal bone height, TT—total bone thickness, CEJ—cementoenamel junction, L1-NB—angle between lower incisor and Nasion-B Point line, IMPA—incisor mandibular plane angle, cl. I—class I malocclusion, LH—lingual bone height, cl. III—class III malocclusion, LT—lingual bone thickness, L1-Apog—distance from lower incisor to A line extending from Point A to pogonion, F-CEJ-MBC—the distance between the CEJ and the facial marginal bone crest, L-CEJ-MBC—the distance between the CEJ and the lingual marginal bone crest, IMPLA angle—angle between the mandibular incisor’s long axis and the LP(lingual plane), BT—buccal bone thickness, ↓—decrease, ↑—increase, →—no change.

## Data Availability

Not applicable.

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
