# Peer review of "Bone Remodeling during Orthodontic Movement of Lower Incisors—Narrative Review"

_ijerph, 2022, doi:10.3390/ijerph192215002_

Round 1
Reviewer 1 Report
Dear Authors,
thank you for submitting your article for review.
The overall English language is correct. Some minor corrections needed.
The article needs some adjustments that you can review on the attached PDF.
The recommendation is to add some referenced regarding bone grafting procedures (Wilko et al).
Best of luck

Author Response
Dear Reviewer,
thank You for Your suggestions. We've expanded the summary and recommendations with more informations about the accelerated osteogenic orthodontics technique with bone graft augumentations. We've added the references regarding the Wilcko at all. studies.
Kind regards,
Anna Grzebyta
Reviewer 2 Report
This review of the literature has been correctly compiled and presented.
No striking new facts could be found, however, with the only exception to carefully monitor the decompensation of lower incisors before orthognathic surgery in class III patients.

Author Response
Dear Reviewer,
Thank You for Your feedback! We appreciate Your comment.
Kind regards,
Anna Grzebyta
Reviewer 3 Report
Thank you for your work on this article.
This article is well organized and has a scientific basis.
I have a some question about your article.
Q1. in line 70. You are referring to the limitations of cephalometric method. But that part goes beyond the limits and feels negative. This method is still being used throughout clinical practice. It would be better to express it as having clinical limitations.
Q2. in summary and recommendtions part, you have simply summarized the overall clinical recommendations based on the included articles. However, it would be even better if you have your own clinical recommendations.
Author Response
Dear Reviewer,
please see the attachment.
Best regards,
Anna Grzebyta

Reviewer 4 Report
Thank you for giving me this opportunity to review the article entitled, " Bone remodelling during orthodontic movement of lower incisors - narrative review"
I carefully reviewed the submitted set of the manuscript and:
- missing an "e" in the word "narrativ" - line 11;
- line 47: "The theory assumes that necrosis occurs ...". I think it is necessary to add that it is an aseptic necrosis.
- why the authors only choose articles that only observed lower teeth? Why didn't add the articles that talk about upper and lower anterior teeth (and only remove the data related to the lower teeth)?
- Table 1 needs some improvements. A list of abbreviations is needed. It is also necessary a header on all pages. It might be a good idea to put the table vertically.
- Throughout the manuscript, when an acronym is written for the first time in the text, it is necessary to write the meaning in parentheses.
- Although the research seems scientifically correct and are submitting the article to a journal with IF=4.614, why did the authors not choose to carry out a systematic review instead of a narrative...?
Author Response

(The authors gave the same response as above.)

Round 2
Reviewer 4 Report
In my opinion, the manuscript is now ready to be accepted